# Population Genomic Evidence for the Diversification of *Bellamya aeruginosa* in Different River Systems in China

**DOI:** 10.3390/biology12010029

**Published:** 2022-12-23

**Authors:** Qianqian Zeng, Yaxian Sun, Hui Zhong, Conghui Yang, Qinbo Qin, Qianhong Gu

**Affiliations:** State Key Laboratory of Developmental Biology of Freshwater Fish, Engineering Research Center of Polyploid Fish Reproduction and Breeding of the State Education Ministry, College of Life Sciences, Hunan Normal University, Changsha 410081, China

**Keywords:** *Bellamya aeruginosa*, SLAF-seq, population genomics, phylogenetics, population structure

## Abstract

**Simple Summary:**

The population genomic study of seven populations of *Bellamya aeruginosa* across three river systems in China was conducted by specific-locus amplified fragment sequencing (SLAF-seq). A clear division was found among populations from the Yellow River basin and the Pearl River basin, as well as population YC from the Yangtze River basin using the SNPs data. However, there existed no distinct population structure using the mitochondrial DNA. Anthropogenic translocation from the Yangtze River to the Pearl River basin and the passive dispersion from the Yangtze River basin to the Yellow River basin by flooding have weakened the phylogeographic pattern of *B. aeruginosa*. These results provide useful guidance for the effective selective breeding of *Bellamya*, which is very important for the development of the industry of LZRSRN.

**Abstract:**

Clarifying the genetic structure can facilitate the understanding of a species evolution history. It is crucial for the management of germplasm resources and providing useful guidance for effective selective breeding. *Bellamya* is an economically and ecologically important freshwater snail for fish, birds and even humans. Population genetic structures of the *Bellamya* species, however, were unknown in previous studies. Population genomics approaches with tens to hundreds of thousands of single nucleotide polymorphisms (SNPs) make it possible to detect previously unidentified structures. The population genomic study of seven populations of *B. aeruginosa* across three river systems (Yellow River, Yangtze River and Pearl River) in China was conducted by SLAF-seq. SLAF-seq obtained a total of 4737 polymorphisms SLAF-tags and 25,999 high-consistency genome-wide SNPs. The population genetic structure showed a clear division among populations from the Yellow River basin (YH and WL) and the Pearl River basin (QSH and LB), as well as population YC from the Yangtze River basin using the SNPs data. However, there existed no distinct population structure using the mitochondrial DNA (mtDNA). The anthropogenic translocation from the Yangtze River basin to the Pearl River basin and the passive dispersion from the Yangtze River basin to the Yellow River basin by flooding have weakened the phylogeographic pattern of *B. aeruginosa*. The divergence of *B. aeruginosa* in the three river systems suggests that the anthropogenic dispersal for aquaculture and breeding requires serious consideration of the population structure for the preservation of genetic diversity and effective utilization of germplasm resources.

## 1. Introduction

The process of population isolation and differentiation is the core of micro-evolution, which has long fascinated evolutionary biology [1,2,3]. The present-day genetic structure and diversity of a species reflect both historical and contemporary patterns of dispersal and gene flow among populations [4]. Freshwater gastropods represent interesting models to study the effects of historical and contemporary factors on the population’s genetic structure due to their primarily sessile lifestyle [5,6].

As one of the most important gastropods in freshwater systems, *Bellamya* has attracted considerable attention from ecological physiologists and evolutionary biologists. *B. aeruginosa* can significantly change the water’s physicochemical properties and the phytoplankton community [7]. *Bellamya* plays an important role in mixing surface sediments and breaking down organic detritus, enhancing microbial growth and nutrient cycling [8,9]. The presence of *B. aeruginosa* can significantly reduce the content of suspended particles and improve water transparency, and even promote the growth of *Vallisneria natan* [10]. Furthermore, the feeding and physiological metabolism of *Bellamya* has significant effects on the suspended particles, nutrients and algal population in the water and can even eliminate the formation of algal blooms [11,12]. They are also widely used in ecotoxicology research as an important indicator for environmental monitoring [13,14,15,16]. Additionally, they are not only the natural diets for fish, crab and even birds but also supply a delicious protein source for humans. It is a very popular snack in China, especially in Guangxi, where “Liuzhou River snail rice noodles” (LZRSRN) is very famous and have been exported to foreign countries and regions [17,18]. The LZRSRN has become a “1.5 million dollar enterprise” in Guangxi [19]. Consequently, as an economically important snail and one of the important materials for the production of LZRSRN, the *Bellamya* was transported not only in large quantities but also over substantial distances [4,17]. The substantial consumption of *Bellamya* was predicted to have a great influence on its genetic diversity and structure. Gittenberger (2012) [20] suggested that anthropogenic translocations and accidental transport may have altered the natural patterns of snails. Snails are supposed to possess strong genetic differentiation between populations or distinct phylogeographical patterns due to their primarily sessile lifestyle. However, the land snails even managed to travel over thousands of kilometers of open ocean and then back again [20], explaining the lack of strong differentiation across large geographic distances. For example, aerial dispersal by birds and anthropogenic translocations also were used to explain the higher intra-population divergence than inter-population of *Isabellaria buresi pharsalica*, and the aerial dispersal by birds explained its occurrence in Thessaly [21]. There existed no significant levels of genetic differentiation among populations, and weak geographic structure was detected despite large geographic distances between populations of *Bellamya* [4,22,23]. Furthermore, recent studies have shown that the species differentiation of *Bellamya* in Africa and China was both indistinct because of interspecific hybridization and gene introgression, and many species should be defined as subspecies [6,24,25]. The *Bellamya* species with indistinguishable shell characteristics from China did not show any clear differentiation and almost formed a single genetic cluster [6]. The divergence time estimation showed that the *Bellamya* in China experienced recent rapid radiation [6,17]. However, the indistinct interspecific relationship and weak geographic structure are averse to its scientific nature conservation and reasonable resource management.

High-throughput Illumina sequencing through the Illumina HiSeq platform has revolutionized molecular ecology [26]. For Illumina sequencing, firstly, DNA strands are fragmented, and then the adapter sequences are attached to the ends of these fragments. These adapters contain the sequence, which can bind to the flow cell, a primer sequence for amplification during sequencing. Indexes can be included in adapters at one or both ends of the DNA fragments. It is required when multiplexing samples within a single sequencing lane. Therefore, the increase in output of a single sequencing lane will commonly result in an increase in multiplexing. For the study of molecular ecology, it is very useful to maximize sample size by using low-coverage whole genome data [26,27].

Using the high-throughput Illumina sequencing of restriction-site associated libraries to generate genome-wide genotypic data and haplotype-tagging SNPs has been particularly valuable for ecological, evolutionary, phylogeographic and genetic mapping studies. They are based on analyzing tens to hundreds of thousands of SNPs in hundreds of barcoded samples at the same time [28]. These data are analyzed using phylogenomic and population genomic approaches to test species boundaries and explore the level of population structure and gene flow [29,30,31]. The mitochondrial topology is susceptible to introgressive hybridization and incomplete lineage sorting and may not adequately reflect the history of these species [32,33]. While nuclear genes are limited by the small number of loci, it is difficult to harbor enough genetic variation in short evolutionary history, particularly for non-model species with a close relationship or rapid radiation [34,35]. The genome-wide genotypic data and haplotype-tagging SNPs are particularly valuable for phylogenetic application and population genomics in non-model species due to the ability to sample a large number of informative polymorphism without genomic resources [36,37]. Compared with traditional Sanger sequencing, high throughput sequencing technologies have gained much attention as they enable rapid generation of up to thousands of loci randomly scattered across the genome and are especially suitable for non-model species [38]. Particularly, it has been successfully used to infer the recent evolutionary history (including diversification, dispersion, isolation, and even hybridization and introgression) of many organisms [39,40,41,42].

*B. aeruginosa* is widely distributed in the Yangtze River basin, Yellow River basin and Pearl River basin. The evolution history of the three drainage systems would play a very different role in the diversification of *B. aeruginosa*. However, previous studies using mtDNA and SSR markers showed that there is nonsignificant genetic differentiation among populations in the three river systems and a weak phylogeographical structure in China [4,6,17,22]. In this study, we sampled seven populations of *B. aeruginosa* distributed over a long distance across the three river systems in China. In order to clarify the population genetic structure of *B. aeruginosa*, we used a population-genomic approach employing SLAF-seq for genotyping thousands of single nucleotide polymorphisms (SNPs) and compared these results with the methods using mitochondrial cytochrome oxidase I (*COI*). SLAF-seq has been widely used as an efficient large-scale genotyping method to identify single nucleotide polymorphisms (SNPs) at low cost due to its reduced representation of deep sequencing without sacrificing the genotyping accuracy [43].

## 2. Materials and Methods

### 2.1. Sample Collection and DNA Extraction

*B. aeruginosa* was collected from different river systems, including the Pearl River basin (two populations), the Yangtze River basin (three populations) and the Yellow River basin (two populations) (Table 1, Figure 1). A total of 68 individuals were obtained from March to September in 2018. Adductor muscles were dissected and preserved in 100% ethanol and then stored at −20 °C until DNA extraction. Genomic DNA was extracted with the DNeasy Blood and Tissue Kit (Tiangen, Beijing, China) following the manufacturer’s protocol, and the extracted DNA was quantified with a spectrophotometer (NanoDrop 2000c, Thermo Fisher Scientific, Waltham, MA, USA).

### 2.2. SLAF-Seq and SNPs Calling

All of the 68 *B. aeruginosa* individuals representing 7 populations (minimum n = 7, maximum n = 12) were used in the SLAF-seq. The SLAF library was constructed according to previous studies [44,45,46]. In the absence of the genome of *Bellamya*, the restriction enzyme combinations were selected in silico based on the related species Oyster (*Ostrea gigas thunberg*) reference genome (557.72 Mb) to optimize the SLAF-seq yields and efficiency. The purified genomic DNA was digested into fragments of 314–364 bp using two restriction enzymes (*RsaI* and *HaeIII*, New England Biolabs, Ipswich, MA, USA). The accuracy of the restriction enzyme digestion protocol was tested according to the control genome (*Oryza sativaindica* genome). Subsequently, SLAF library was constructed with the selected DNA fragments followed by fragment end reparation, index paired-end adapters’ ligation and adapter-modified ends, PCR amplification, and target fragment selection. High-throughput sequencing was performed with the 100 bp paired-end method on the Illumina HiSeqTM 2500 (Illumina, Inc.; San Diego, CA, USA) at the Biomarker Technologies Corporation in Beijing. The ratio of high-quality reads with quality scores greater than Q30 and guanine-cytosine (GC) content in raw reads was estimated for quality control. We also used the Seqtk (https://github.com/lh3/seqtk, accessed on 18 June 2018) to filter the raw reads with low-quality bases or adaptor/primer contamination. Next, we mapped the high-quality paired end reads onto the reference genome (*O. gigas thunberg*) using the SOAP2, according to Li et al. (2009) [47]. Subsequently, the SAM files were converted into indexed BAM files using SAMtools v1.5 [48]. The low-quality SNPs (QUAL < 30.0; QD < 2.0; FS > 60.0; MQ < 40.0; and SOR > 3.0) were removed, and the high-quality SNPs were identified by the software GATK v3.8 [49]. The final dataset contained 68 individuals, and 25,999 unlinked SNPs were selected with the missing rate at the marker level set at ≤50%, minor allele frequency (MAF) above 0.05, and integrity >0.80 for further analyses.

### 2.3. Phylogenetic Analysis and Population Clustering Analyses

We generated genomic phylogeny using an unrooted neighbor-joining (NJ) tree with 100 bootstraps in SNPhylo [38], which has been widely used in population genetics and phylogenomics [50]. We used the FIGTREE v1.4.4 [51] to visualize and format the phylogenetic tree. We tested population structure using the block relaxation approach assignment test in the program ADMIXTURE v1.3 [52] to estimate the ancestry of individuals, with testing cluster values of *K* = 1 to *K* = 10 [53]. To determine the most appropriate value of *K*, we conducted a 10-fold cross-validation and selected the estimate with the lowest error, and the output was visualized in the R package pophelper. The CLUSTER [54] was used to conduct a principal components analysis (PCA), and the number of informative eigenvalues was also used to infer population structure [46]. ARLEQUIN v3.5.2.2 [55] was used to estimate the pairwise population genetic differentiation (*F*_ST_) of *B. aeruginosa*, and 10,000 permutations were used to test the significance (*p*-value < 0.05).

### 2.4. Mitochondrial COI Sequencing and Analysis

The *COI* sequences were amplified using the universal primers for invertebrates described by Folmer et al. (1994) [56]. Next, the PCR products were purified using the Qiagen MinElute PCR Purification Kit (Qiagen Ltd., Manchester, UK), and processed by Sangon Biotech Co., Ltd. (Shanghai, China). All individuals were sequenced in both directions. Forward and reverse sequences were checked and aligned using the SEQMAN software in Lasergene v. 7.1 (DNAstar, Madison, WI, USA). Consensus sequences were blasted, and all new *COI* sequences were deposited in GenBank (OP889150–OP889217).

The resulting 68 sequences were aligned by using MAFFT implemented in PhyloSuite v1.2.2 [57]. The maximum likelihood (ML) method and Bayesian inferences (BI) were used to construct the phylogenetic tree based on the aligned *COI* sequences. ML tree was implemented in RAxML v8.2.4 [58] with 1000 bootstrap replicates. The BI tree was performed in MrBayes v3.2.6 [59] using the Markov Chain Monte Carlo method with 10 million generations and sampling trees every 1000 generations. The first 25% of trees were discarded as burn-in, with the remaining trees being used for generating a consensus tree. The final trees were visualized in FIGTREE v1.4.4.

An analysis of molecular variance (AMOVA) was conducted in ARLEQUIN v3.5.2.2 to quantify the genetic variation of the populations at three different hierarchical levels: among different groups according to three river systems, among populations within different groups, and within populations. The significance level (*p*-value < 0.05) for the variance components was computed via 1000 permutations. The hierarchy for this analysis was selected based on the population of the river systems: (i) QSH and LB; (ii) YW, LZH, and YC; (iii) YH and WL. Furthermore, we analyzed another hierarchical population structure, where individuals were grouped into three clusters in ADMIXTURE. Pairwise *Φ*_ST_ values were calculated to assess population differentiation in ARLEQUIN v3.5.2.2.

## 3. Results

### 3.1. Characterization of SLAF-Seq Data and SNP Identification

After filtering, a total of 23.18 million high-quality paired-end reads were generated from SLAF-seq of 68 *B. aeruginosa* accessions. The average high-quality base ratio (Q30) of all 68 accessions was 90.37%, and the average GC content was 38.86% (Appendix A). In addition, 0.11 M reads were obtained from the control O. sativaindica, which was used to evaluate the accuracy of the established library. After clustering high-quality reads, a total of 19,221 SLAF tags were developed from all accessions, and the average sequencing depth of the SLAF tags was 13.59×. Among the SLAF tags, 4737 were polymorphic (Appendix A). The GATK and SAMtools were used to develop high-consistency SNPs. A total of 25,999 highly consistent SNPs (MAF > 0.05) were identified for subsequent genetic structure analysis.

### 3.2. Population Genetic Structure

In order to ascertain the divergence of *B. aeruginosa* across three river systems, we performed a principal component analysis based on the SNPs data. PCA showed that *B. aeruginosa* in the Yellow River basin and the Pearl River basin were clearly distinguished, but *B. aeruginosa* in the Yangtze River basin was mixed with the other two river systems (Figure 2a). However, all individuals from population YC in the Yangtze River basin clustered into a single group (Figure 2a). The population clustering using admixture proportions from 1 to 10 was performed by the ADMIXTURE program. The result of population clustering indicated that a value of *K* = 3 was associated with the lowest cross-validation procedure (Figure 2b), and the admixture coefficients for each accession at *K* = 3 were listed in Appendix A. The results of *K* = 1, 2, 4–10 were provided in Appendix A. The 68 accessions were attributed to three obvious groups (Group I, II and III) with an admixture coefficient of more than 80%. Group I and III were almost composed of all populations from the Pearl River basin and the Yellow River basin, respectively (Figure 2c), while Group II included only one population, YC, from the Yangtze River basin. Furthermore, eight accessions from YW (Yangtze River basin) were mixed in Group I with an admixture coefficient of more than 99% (Appendix A); three accessions from YW and all accessions from LZH (Yangtze River basin) were related to Group III. All these results indicated a strong divergence between different *B. aeruginosa* populations.

### 3.3. Population Genetic Differentiation

Pairwise *F*_ST_ among populations estimates from the SNPs data range from 0.535 to 0.800 (Table 2); while *Φ*_ST_ estimates from the *COI* range from 0.004 to 0.927. All *F*_ST_ values were high and statistically significant (*p* < 0.05), but most *Φ*_ST_ were low and not statistically significant (*p* > 0.05). Furthermore, pairwise *F*_ST_ and *Φ*_ST_ values between YC and other populations were much higher (Table 2), implying a high differentiation between YC and all other populations. AMOVA revealed that low differentiation at the group level when using river systems as groups. The genetic variance among groups was 12.39% without significance (*p* = 0.11), while 61.16% (*p* < 0.00) of genetic variation was explained within population variation (Table 3). Another AMOVA was conducted according to three clusters in ADMIXTURE, populations among the groups contributed to 27.49% (*p* > 0.05) of the total genetic variance, and most genetic variation was found within the population (Table 3).

### 3.4. Phylogenetic Trees

Phylogenetic analysis showed that the 68 *B. aeruginosa* across three river systems were clustered into three distinct branches based on SNPs data (Figure 3a). Population YC from the Yangtze River basin was clustered into one single branch. LB, QSH, eight individuals from YW, and only one in WL were clustered into another branch. The remaining populations YH, WL and LZH, as well as three individuals from YW were clustered into the third branch. However, individuals from different populations formed a nested structure in the ML tree and BI tree according to the mtDNA *COI* data, and populations from the same river system were not phylogenetically clustered (Figure 3b).

## 4. Discussion

*Bellamya* is an important freshwater snail and is widely distributed in lakes and rivers. It has high nutritional value, a high percentage of delicious amino acids, and is the main source of the famous food LZRSRN [17,18,19]. There have been appreciable advances in the reproduction and cultivation of *Bellamya* in Guangxi, especially in Liuzhou City [18]. However, the key technology of large-scale artificial reproduction for *Bellamya* has not made great progress in practical applications. Currently, the supply of *Bellamya* in LZRSRN continues to depend mainly on natural resources in lakes and rivers. However, a ten-year fishing ban plan on the whole Yangtze River basin was conducted in 2020, and the supply of wild *Bellamya* from the Yangtze River basin has been shut down [18]. Hence, the breeding and artificial culture of *Bellamya* is becoming urgent and necessary. Deciphering the population’s genetic structure provides a scientific basis for the protection of natural germplasm resources and breeding [60,61]. However, the inference of phylogenetic relationships and population genetic structure among recently diverged and non-model species is a challenging problem due to inter-population gene flow or incomplete lineage sorting [36].

In previous studies, the mitochondrial sequencing and microsatellite data, which were neutral molecular markers and widely used in population genetics [62], however, had a low contribution to the identification of the population structure of the *Bellamya* species with low dispersal ability and sessile lifestyle [4,22,23]. They were mainly interpreted by moderate to high level of gene flow among populations caused by anthropogenic translocations, flooding events and even accidental waterfowl dispersal [20,63,64,65]. In the present study, there existed no phylogeographical pattern or population structure for *B. aeruginosa* using mtDNA *COI*, even with the seven populations distributed across different river systems. However, distinctive phylogeographic patterns and three groups of *B. aeruginosa* across the three river systems were found using SLAF-seq; one was in the Yellow River basin, one in the Pearl River basin, and another in the Yangtze River basin. However, the admixture structure was found both between the Pearl River basin and the Yangtze River basin and between the Yellow River basin and the Yangtze River basin. The formation of the Nanling Mountain Range (Figure 1) between the Yangtze River basin and the Pearl River basin would have driven the separation of many freshwater organisms [66]. Previous molecular phylogenetic analyses of fish (*Hemiculter leucisclus*), amphibians (*Andrias sligoi*), and aquatic insects (*Microvelia horvathi*) [67,68,69], which were investigated using mitochondrial genes and microsatellites, have identified genetically distinct local populations and phylogeographic patterns. According to previous divergence dating analyses [6,17], the origin and diversification of *Bellamya* in China were much later than the formation of the Nanling Mountains and the watershed between the Yangtze River basin and the Pearl River basin [70]. The Nanling-Wuyi Mountain range (Figure 1) should represent a major phylogeographic barrier for *B. aeruginosa*. The passive transport of waterfowl across such a huge geographical barrier seems impossible, as the retention time of intact invertebrate propagules in the intestine of a bird is limited [63]. The flooding also cannot explain the strong admixture among far distant sampling locations across the two isolated drainage systems [4]. Consequently, the anthropogenic translocation becomes the main intelligible reason for the gene flow between the Yangtze River and Pearl River basin, and it was reported that the *Bellamya* in LZRSRN had been mainly captured from the lakes distributed in the Yangtze River basin [18], such as Honghu Lake in the Hubei province, Dongting Lake in the Hunan province, and Poyang Lake in the Jiangxi province. Substantial anthropogenic translocation of *Bellamya* from the Yangtze River basin to the Pearl River basin might have weakened the population’s genetic structure.

The admixture between the Yangtze River basin and the Yellow River basin could be attributed to their spatial proximity, which enables *B. aeruginosa* populations from the middle and lower Yangtze River to expand to nearby rivers easily by flooding in summer [67]. There existed no huge geographical barrier between the middle and lower Yangtze River basin and the Yellow River basin (Figure 1). The strong admixture of fish *H. leucisculus* among the middle and lower Yangtze River, Huaihe River and the Yellow River basin was also mainly attributed to the flooding [66].

The seven populations of *B. aeruginosa* received maximal support in the SLAF-seq tree (Figure 3a) and were grouped into three main clades, suggesting that there existed a distinct population structure for *B. aeruginosa* across the three river systems. All the populations from the Pearl River basin and the Yellow River basin clustered into one clade, respectively. Even most individuals from the same populations clustered together, such as the population of QSH, LB, WL, LZH, and especially YC. These results suggested that the population genetic structure of *B. aeruginosa* has not been completely changed by recent human-mediated translocations and flooding. Although there was a weak phylogeographic pattern for *B. aeruginosa* between the Yangtze River basin and Pearl River basin, as well as between the Yellow River basin and Yangtze River basin, YC in the Yangtze River basin formed one single branch. This suggested that *B. aeruginosa* in the Yangtze River basin had a more complex evolution history compared with that in the other two river basins. These findings provide valuable insights into the further characterization of the population genetics of this economically and ecologically important freshwater snail. Further research of population genomics should be executed for *B. aeruginosa* with comprehensive and extensive samples in the future.

Compared with SLAF-seq, the mtDNA *COI* on their own did not support the three clades of *B. aeruginosa,* and the inter-population relationships were not fully resolved in the ML tree and BI tree (Figure 3b). The SLAF-seq approach, however, yielded an almost resolved tree (Figure 3a) with maximum support nodes confirmed. The unidentified population structure of *B. aeruginosa* for mtDNA and nDNA (microsatellite markers) in a previous study [4] could be successfully resolved with SLAF-seq. Hence, the increased number of markers of the SLAF-seq data helped to resolve the population genetic structure of *Bellamya*, coinciding with other recent studies using reduced representation genomic data and population genomic approaches to resolve phylogenetic relationships and population structure for closely related species and non-model species [36,37,41,42,44,71].

## 5. Conclusions

In the present study, 25,999 SNPs and 4737 polymorphisms SLAF-tags were identified using SLAF-seq to investigate the population structure of *B. aeruginosa* across three river systems in China. Compared with the mtDNA *COI* data, 25,999 SNPs revealed that the seven *B. aeruginosa* populations could be divided into three groups with distinct phylogeographic patterns, indicating the power of the SLAF-seq in population genomics for snail. PCA and admixture structure showed that *B. aeruginosa* in the Yellow River basin and Pearl River basin were clearly distinguished, but *B. aeruginosa* in the Yangtze River basin was mixed with the other two river systems. Pairwise *F*_ST_ using SNPs among populations was much higher and statistically significant, suggesting a high population differentiation, while most pair *Φ*_ST_ values were lower, except between population YC and other populations. The result of the phylogenetic analyses almost coincided with PCA and admixture structure results, where population QSH and LB from the Pearl River basin clustered together with some individuals from YW, which belongs to the Yangtze River basin; population YH and WL from the Yellow River basin clustered together with LZH and several individuals from YW; the remaining population YC from the Yangtze River basin clustered as a single clade. The previously unidentified population structure of *B. aeruginosa* was almost resolved by SLAF-seq, providing useful guidance for effective selective breeding and further conservation strategies for *Bellamya* germplasm resources, as the LZRSRN is making robust progress.

## Figures and Tables

**Figure 1 biology-12-00029-f001:**
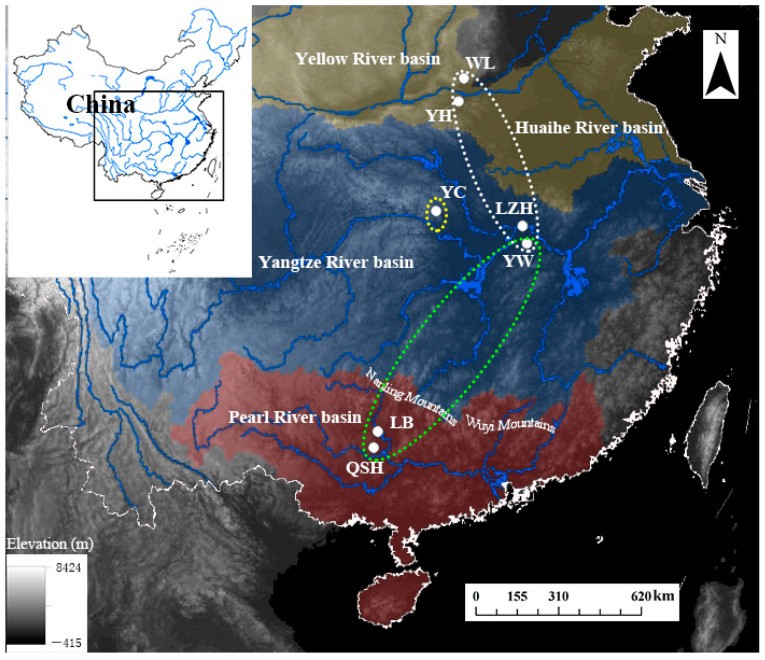
Sampling map with population codes of *B. aeruginosa* (Table 1) across three river systems in China: two populations (YH and WL) in the Yellow River basin (yellow area), three populations (YC, LZH, YW) in the Yangtze River basin (blue area), and two populations (LB and QSH) in the Pearl River basin (red area).

**Figure 2 biology-12-00029-f002:**
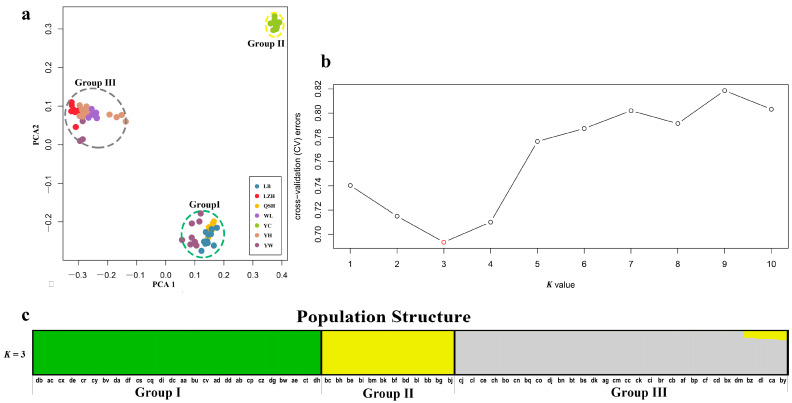
(**a**) Plot of principle component analysis (PCA). The seven populations of *B. aeruginosa* were divided into 3 groups, the color of each accession is indicated by the population it belongs to, and Group I (including LB, QSH and YW), II (including YC) and III (YH, WL, LZH and YW) were represented by green, yellow and gray dotted ellipse, respectively. (**b**) Threefold cross-validation (CV) error values across different values of K in ADMIXTURE. (**c**) Population structure analysis based on 68 accessions of *B. aeruginosa* using ADMIXTURE with the optimal clustering number set at *K* = 3. Each accession was indicated by a vertical column, and the colored portion (green, yellow, and gray) in each column represents the proportion contributed from ancestral populations. The accession code was placed at the bottom according to BMK ID in Appendix A.

**Figure 3 biology-12-00029-f003:**
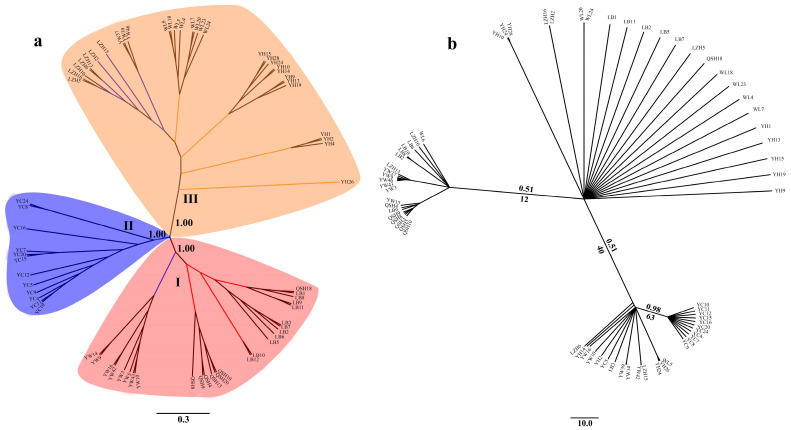
Phylogenetic tree of 68 accessions of *B. aeruginosa*. (**a**) Neighbor-joining (NJ) tree based on SNPs; the confidence coefficient is shown on the branch. (**b**) The Bayesian inferences (BI) tree and maximum-likelihood (ML) tree based on mtDNA *COI* (the same individuals as SLAF-seq); numbers at nodes are Bayesian posterior probabilities (up the branch) and maximum likelihood bootstrap values (down the branch).

**Table 1 biology-12-00029-t001:** The sample information of seven populations of *B. aeruginosa* included in this study, n means the number of individuals in each population. CN means China.

Pop ID	Location	River System	Longitude	Latitude	n
YH	Luoyang, Henan, CN	Yellow River	112.438594	34.402625	12
WL	Jiyuan, Henan, CN	Yellow River	112.696461	35.160814	8
YW	Xianning, Hubei, CN	Yangtze River	114.657836	29.880284	11
LZH	Wuhan, Hubei, CN	Yangtze River	114.535902	30.423833	7
YC	Yichang, Hubei, CN	Yangtze River	111.438069	30.945422	12
QSH	Laibing, Guangxi, CN	Pearl River	109.186071	23.761714	7
LB	Liuzhou, Guangxi, CN	Pearl River	109.322252	24.259994	11

**Table 2 biology-12-00029-t002:** Pairwise *F*_ST_ (below diagonal) and *Φ*_ST_ (above diagonal) among populations of *B. aeruginosa*. * means significant *p*-values < 0.05.

Pop	YH	WL	YW	LZH	YC	QSH	LB
YH		−0.030	0.130	0.024	0.577 *	0.305 *	0.091
WL	0.536 *		0.193	−0.004	0.769 *	0.338 *	0.030
YW	0.563 *	0.587 *		0.090	0.444 *	0.316 *	0.231
LZH	0.589 *	0.635 *	0.591 *		0.730 *	0.279	0.029
YC	0.729 *	0.771 *	0.700 *	0.800 *		0.926 *	0.800 *
QSH	0.645 *	0.681 *	0.564 *	0.713 *	0.744 *		0.309
LB	0.541 *	0.644 *	0.535 *	0.677 *	0.706 *	0.541 *	

**Table 3 biology-12-00029-t003:** Analysis of molecular variance (AMOVA) results comparing genetic variation in *B. aeruginosa* collected from 7 populations across three river systems. * means significant *p*-values < 0.05.

	*df*	Sum of Squares	Variance of Components	Percentage	Φ-Statistics	*p* Value
River systems						
Among groups	2	66.422	0.68510 Va	12.39	*Φ*_CT_ = 0.12393	0.109
Among populations with groups	4	68622	1.46189 Vb	26.44	*Φ*_SC_ = 0.30185 *	0.000
Within populations	61	206.251	3.38116 Vc	61.16	*Φ*_ST_ = 0.38837 *	0.000
Admixture						
Among groups	2	92.028	1.58826 Va	27.49	*Φ*_CT_ = 0.27485	0.053
Among populations with groups	4	43.016	0.80916 Vb	14	*Φ*_SC_ = 0.19310 *	0.002
Within populations	61	206.251	3.38116 Vc	58.51	*Φ*_ST_ = 0.41488 *	0.000

## Data Availability

Not applicable.

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
