# Peer review of "Population Genomic Evidence for the Diversification of Bellamya aeruginosa in Different River Systems in China"

_biology, 2022, doi:10.3390/biology12010029_

Round 1
Reviewer 1 Report
The study on ‘Population genomic evidence for the diversification of Bellamya aeruginosa in different River system in China’ by Zeng et al., has focused on the anthropogenic dispersal for aquaculture and breeding requires solemn consideration on the population structure of gastropods for the preservation of genetic diversity and effective utilization of germplasm resources. To me, it’s an interesting study and the MS is well-written. However, the manuscript needs some minor improvements before considering for final acceptance:
1. Page-2, line 50-51: “B. aeruginosa can significantly change the water physicochemical properties and phytoplankton community” – need more explanation and use more references.
2. Page-2, line 53-54: need more citations.
3. Page-2, line 61-64: “Gittenberger (2012) [14] suggested that anthropogenic translocations and accidental transport may have altered natural patterns of snails.”-how? please explain.
4. Page 2, line 64-66: ‘Previous studies have interpreted these results as evidence for long-distance dispersal of freshwater snail with low dispersal ability and sessile life style.” I suggest, this sentence should be moved to discussion part.
5. Please add a paragraph on “high-throughput Illumina” in the introduction part.
6. Page 3, line 108-111: Please consider to delete this sentence. And add it in the conclusion part.
7. How did you select the study areas and why?
9. What is the specific conclusion of the study?
Overall, good organized MS and enjoyed reading the contents.
Author Response
Reviewer 1
The study on ‘Population genomic evidence for the diversification of Bellamya aeruginosa in different River system in China’ by Zeng et al., has focused on the anthropogenic dispersal for aquaculture and breeding requires solemn consideration on the population structure of gastropods for the preservation of genetic diversity and effective utilization of germplasm resources. To me, it’s an interesting study and the MS is well-written. However, the manuscript needs some minor improvements before considering for final acceptance:
Response:
Thanks for your comment and good constructive suggestion for this article.
- Page-2, line 50-51: “B. aeruginosacan significantly change the water physicochemical properties and phytoplankton community” – need more explanation and use more references.
Response:
Thanks for your suggestion, more explanations and references for this sentence were supplemented in the revised manuscript. Please see line 50-54 in Page-2.
- Page-2, line 53-54: need more citations.
Response:
Thanks for your suggestion, more citations have been included.
- Page-2, line 61-64: “Gittenberger (2012) [14] suggested that anthropogenic translocations and accidental transport may have altered natural patterns of snails.”-how? please explain.
Response:
Thanks for your suggestion, the explanation for this has been added in the revised manuscript. Please see line 68-72 in Page-2.
- Page 2, line 64-66: ‘Previous studies have interpreted these results as evidence for long-distance dispersal of freshwater snail with low dispersal ability and sessile life style.” I suggest, this sentence should be moved to discussion part.
Response:
Thanks for your suggestion, changed accordingly.
- Please add a paragraph on “high-throughput Illumina” in the introduction part.
Response:
Thanks for your suggestion. We added a paragraph on “high-throughput Illumina” in the introduction part. Please see line 86-95 the Page-2.
- Page 3, line 108-111: Please consider to delete this sentence. And add it in the conclusion part.
Response:
Thanks for your suggestion. This sentence has been deleted and added in the conclusion part.
- How did you select the study areas and why?
Response:
Thanks for your comment. We added the reason for selecting the study area in the last part of Introduction (line 116-121) as following: aeruginosa is widely distributed in Yangtze River basin, Yellow River basin and Pearl River basin, which had very different evolution histories. The formation of the three drainage systems would play a very different role on the diversification of B. aeruginosa. Previous studies using mtDNA and SSR markers showed that there is nonsignificant genetic differentiation among populations in the three river systems. Hence, we use a population genomic to study the diversification among populations distributed with long distance in the three river systems.
8. What is the specific conclusion of the study?
Response:
The distinct population structure and phylogegraphical pattern of B. aeruginosa can be clarified by 25,999 SNPs identified using SLAF-seq, but not by mtDNA COI, indicating the power of the SLAF-seq in population genomics for snail.

Reviewer 2 Report
This is an excellent manuscript that assess the genetic structure of the freshwater snail Bellamya aeruginosa in China, based on a population genomic study using SNPs, and comparing the results with the information provided by mitochondrial DNA. The MS is in very good shape, the writing is clear, the methodology is sufficiently detailed, and the results obtained are consistent with the methodology employed. Also the figures are high quality. I have not found details that need to be corrected, so I congratulate the authors for such a nice MS.
Author Response
Thanks for your comments on the manuscript very much.
Reviewer 3 Report
Mention the company details of all instruments used in this study in the “material and method” section.
• It is better to add the registration number of the ethical review board if available.
• The standard “P-value” is not mentioned in the “Statistical analysis”.
• The “significant” indicators are not mentioned in all figures. Mention it to show that the difference is significant or not compared with the control.
• The results need minor revision and must be presented well mannered.
• There are several typo mistakes throughout the manuscript. These must be corrected.
• Carefully check the citation and bibliography in the manuscript. The references are not according to the journal style.
• The conclusion section is too short must be revised. Add the findings of ypur study in it briefly.
• The language of the manuscript must be revised.
• It is noted that several sentences in this study are without evidence (references). Provide the latest references in each literature claim both in the “introduction” and in the “dicussion”.
• It is better to add some references in the “material and method” sections to support the methods from the other published research.
Author Response
Comments and Suggestions for Authors
- Mention the company details of all instruments used in this study in the “material and method” section.
Response:
Thanks for your suggestion, all the company details of all instruments have been mentioned in the revised manuscript.
- It is better to add the registration number of the ethical review board if available.
Response:
Thanks for your suggestion. The ethical approval is not applicable in this study.
- The standard “P-value” is not mentioned in the “Statistical analysis”.
Response:
Thanks for your careful checking. The standard “P-value” for significance level has been mentioned in the revised manuscript, please see line 183, 203.
- The “significant” indicators are not mentioned in all figures. Mention it to show that the difference is significant or not compared with the control.
Response:
Thanks for your careful checking. The “P-value” for significance level has been mentioned in the Table 2 and Table 3.
- The results need minor revision and must be presented well mannered.
Response:
Thanks for your suggestion. The results have been revised and the titles presented consistently.
- There are several typo mistakes throughout the manuscript. These must be corrected.
Response:
Thanks for your careful checking. All the typo mistakes have been corrected.
- Carefully check the citation and bibliography in the manuscript. The references are not according to the journal style.
Response:
Thanks for your careful checking. We have carefully checked the citation and bibliography in the revised manuscript.
- The conclusion section is too short must be revised. Add the findings of your study in it briefly.
Response:
Thanks for your suggestion. We added the main findings in the conclusion in the revised manuscript, please see line 371-378.
- The language of the manuscript must be revised.
Response:
Thanks for your suggestion. The language has been improved by who has spent more than four years in the France and hence is fluent in English. He has checked and revised the whole manuscript.
- It is noted that several sentences in this study are without evidence (references). Provide the latest references in each literature claim both in the “introduction” and in the “dicussion”.
Response:
Thanks for your good constructive suggestion. The references have been updated in the “introduction and discussion”.
- It is better to add some references in the “material and method” sections to support the methods from the other published research.
Response:
Thanks for your constructive suggestion. Some references have been added in the “material and method” for the methods published in other researches. Please see line 149, 174, 177, 181.
